# Design of a Double-Layer Electrothermal MEMS Safety and Arming Device with a Bistable Mechanism

**DOI:** 10.3390/mi13071076

**Published:** 2022-07-07

**Authors:** Kexin Wang, Tengjiang Hu, Yulong Zhao, Wei Ren, Jiakai Liu

**Affiliations:** 1State Key Laboratory for Manufacturing Systems Engineering, Xi’an Jiaotong University, Xi’an 710049, China; wkx741565@stu.xjtu.edu.cn (K.W.); zhaoyulong@xjtu.edu.cn (Y.Z.); 2Science and Technology on Applied Physical Chemistry Laboratory, Shaanxi Applied Physical Chemistry Research Institute, Xi’an 710061, China; rw0192@163.com; 3School of Management and Assurance, Armed Police Engineering University, Xi’an 710086, China; liujiakai1129@163.com

**Keywords:** MEMS, pyrotechnics, safety and arming device, electro-thermal actuator, bistable mechanism

## Abstract

Considering the safety of ammunition, safety and arming (S&A) devices are usually designed in pyrotechnics to control energy transfer through a movable barrier mechanism. To achieve both intelligence and miniaturization, electrothermal actuators are used in MEMS S&A devices, which can drive the barrier to an arming position actively. However, only when the actuators’ energy input is continuous can the barrier be stably kept in the arming position to wait for ignition. Here, we propose the design and characterization of a double-layer electrothermal MEMS S&A Device with a bistable mechanism. The S&A device has a double-layer structure and four groups of bistable mechanisms. Each bistable mechanism consists of two V-shape electrothermal actuators to drive a semi-circular barrier and a pawl, respectively, and control their engagement according to a specific operation sequence. Then, the barrier can be kept in the safety or the arming position without energy input. To improve the device’s reliability, the four groups of bistable mechanisms are axisymmetrically placed in two layers to constitute a double-layer barrier structure. The test results show that the S&A device can use constant-voltage driving or the capacitor–discharge driving to drive the double-layer barrier to the safety or the arming position and keep it on the position passively by the bistable mechanism.

## 1. Introduction

Safety and arming (S&A) devices are the core component of pyrotechnics to control the energy transfer (allowing normal ignition and preventing accidental ignition) through a movable barrier mechanism. Therefore, its performance is the key to ensure the safety, reliability, and damage efficiency of the weapon system. Following the increasing demand for weapon performance in modern battle, the MEMS S&A devices have become a research focus due to their outstanding characteristics of micro-mechanism, integrated structure, and intelligent driving [1,2]. Classified by the barrier’s driving principle, the MEMS S&A devices mainly include inertial [3,4,5,6,7,8], electrothermal [9,10], electromagnetic [11,12,13,14], and pyrotechnic [15,16]. Compared with other principles, the electrothermal S&A devices are simpler in structure and gain the active control of the pyrotechnics’ condition.

In existing electrothermal S&A devices [9,10], the barrier connected to the actuator directly shows a faster response and constitutes a more integrated structure [10]. When a voltage is applied, the actuator will be heated and generate thermal expansion to drive the barrier into the arming position. To be ignited at any time, the pyrotechnics require that the barrier must be kept in the arming position stably. Therefore, a continuously applied voltage is needed to maintain the displacement of the barrier. However, heating the actuator continuously will increase the overall temperature of the device, which is also a waste of energy for the weapon system. Thus, the electrothermal S&A devices need a bistable mechanism to keep the barrier in the safety or the arming position passively. In existing research, the bistable function is usually realized by a bistable beam and two electrothermal actuators, and these bistable mechanisms are mainly applied on the bistable switches [17,18,19,20]. The switches only need a small displacement to be disconnected and focus on the contact resistance to improve switching performance. However, the S&A device needs a large displacement and forces to fix the barrier on safety or arming position, and not concerned with the electrical performance. Therefore, applying the bistable mechanism to the S&A device needs more exploration.

Herein, we propose the design and characterization of a double-layer electrothermal MEMS S&A Device with a bistable mechanism, which has a double layer structure and four groups of bistable mechanisms. Each bistable mechanism consists of two V-shape electrothermal actuators to drive a semi-circular barrier and a pawl, respectively, and control their engagement according to a specific operation sequence. Then, the barrier can be kept in the safety or the arming position without energy input. The test results show that the S&A device can use constant-voltage driving or the capacitor–discharge driving to drive the double-layer barrier to the safety or the arming position and keep it on the position passively by the bistable mechanism.

## 2. Modeling

### 2.1. Fundamental Mechanism of the S&A Device

The S&A device consists of two layers (top layer and bottom layer) of SOI wafers (50 μm device layer, 3 μm buried layer, and 400 μm handle layer), and each layer has four V-shape electro-thermal actuators (two pawl actuators and two barrier actuators), as shown in Figure 1a. All the eight actuators in two layers constitute four groups of bistable mechanisms, and each bistable mechanism has a semi-circular barrier and a pawl, which are axisymmetrically arranged in the center of the S&A device to form a double-layer barrier with a total thickness of 100 μm to enhance structure strength. The S&A device integrates a double-layer barrier and four groups of the bistable mechanisms within an overall size of 6.8 × 6.8 × 0.9 mm^3^, as shown in Figure 1b. The actuators on the top and bottom layers are connected to electrodes exposed at the corners of the assembled S&A device (Figure 1b). The section view of the S&A device is shown in Figure 1c. To create a chamber, the handle layer is protruding, the barrier in the device layer is sunken, and the top layer is reversed, covering the bottom layer. The diameter of the top chamber connecting the energetic materials is 1 mm, and that of the bottom one as the fire hole to connect the detonator is 0.5 mm. The double-layer barrier is able to completely cover the fire hole (0.5 mm) with a total diameter of 0.56 mm.

The mission of the S&A device is to allow normal detonation and prevent accidental detonation. When the S&A device is in the safety condition, the barrier will cover the fire hole and allow no signal in. If the detonator is accidentally ignited, explosive energy will be kept within the fire hole and cannot reach and detonate the energetic materials, as shown in Figure 1d. Otherwise, driven by an arming signal, the fire hole will be open and the S&A device will be switched to the arming condition. Explosive energy will be allowed to pass through the S&A device and detonate the energetic materials, as shown in Figure 1e.

To realize the bistable function, only according to a specific operation process can the S&A device switch the conditions to the safety or the arming. The operation process is shown in Figure 1f: (1) Opening the pawls to release the barriers’ movement; (2) Opening the barriers to the arming position; (3) Closing the pawls to prevent the barriers from recovering to the safety position; (4) Closing the barriers to engage with the pawls; (5) The pawls and the barriers had engaged at the arming position, and the S&A device had switched to the arming condition completely. According to a reverse operation process, the S&A device will recover to the safety condition. Thus, the bistable mechanism can keep the double-layer barrier in the safety or the arming position passively based on this principle. The specific operation process can also increase the safety of the S&A device.

### 2.2. Driving Principle of the S&A Device

All the pawls and the barriers are driven by V-shape electro-thermal actuators. Their performance directly affects the safety and arming function of the weapon system. The basic unit of the actuator is a V-shape silicon beam, which can generate thermal expansion with current flowing through. According to previous research [9,10], the temperature distribution of the beam based on one-dimensional heat diffusion model can be calculated by applying the following equations:(1)$$k_{s}\frac{d^{2}T(x)}{dx^{2}}+J^{2}\rho =\frac{S}{h}\frac{T(x)-T_{r}}{R_{T}}$$

With the thermal boundary conditions considered:(2)$$T(0)=T(L)=T_{r}$$
where *k_s_* is the thermal conductivity of silicon. *J* represents the electrical current density, *J* = *V*cos*θ*/*ρL*, for which *V* is the voltage applied on the actuator, and *ρ* is the electrical resistivity of silicon. *S* is the shape factor of the actuators, *T_r_* is the reference temperature, which is equal to the room temperature, and *R_T_* is the thermal resistance of the actuator’s surface. Other geometrical parameters of the actuators are shown in Table 1 and presented in Figure 2. To enhance the performance of the actuator, two chambers are designed on the upper and lower surface of the actuator to increase thermal resistance.

Based on the heat diffusion model, the thermal expansion of the V-shape silicon beam can be obtained. However, as the thermal expansion coefficient of silicon is so small, the actuator is hard to drive the barrier directly. Therefore, the actuator needs a soft lever mechanism to enlarge the deformation. The soft lever mechanism consists of a hard lever and two soft beams, and the barrier is placed on the end of the hard lever. One soft beam is fixed and the other one is connected to the actuator. The displacement of the barrier is enlarged due to the bending deformation of the soft silicon beam. The balance between the actuator’s out force and the soft lever’s driven force can be obtained by applying the following equations [9,10]:(3)$$\left\{\begin{array}{l} \tan(\frac{L_{d}L^{\prime }k_{y}(d^{\prime }-d)}{EI})=\frac{d}{L_{d}}\\ k_{y}=\frac{4\sin^{2}\theta \cdot AE\cos\theta }{L}\\ d^{\prime }=\frac{L}{2}(\sqrt{\frac{{\left(1+\alpha (T-T_{r})\right)}^{2}}{\cos^{2}\theta }-1}-\tan\theta ) \end{array}\right.$$
where *d* and *d’* is the displacement of the actuator with soft lever and without soft lever, *k_y_* is the elastic coefficient of the actuator, *E* is the Young’s modulus of silicon, *I* is the moment of inertia, *I* = *hw’*^3^/12, *A* is the cross-sectional area of the actuator, *α* is the thermal expansion coefficient of silicon, and *T* is the average temperature of the beam. The specific geometrical parameters of the soft lever mechanism are shown in Table 2 and presented in Figure 2. According to calculations, when the applied voltage (*V*) is set to 12 V, the displacement of the barrier actuators is 9.03 μm without the soft lever mechanism, and the maximum temperature is 1143 K. After integrating the soft lever mechanism, the displacement reduces from 9.03 μm to 7.75 μm, and the barrier’s displacement is enlarged to 367.5 μm. Similarly, the displacement of the pawl actuators reduces from 8.17 μm to 6.42 μm, and the maximum temperature is 1144 K. After being enlarged, the pawl’s displacement is 149.4 μm.

Finite element simulations of the actuators are carried out, and the results are shown in Figure 2. When the applied voltage is set to 12 V, the barrier actuators and the pawls actuator can generate a displacement of 7.59 μm and 6.41 μm. After being enlarged by the soft lever mechanism, the displacement of the barrier (*d_B_*) is 325.1 μm (error of 11.5%), which enables the ignition chamber (250 μm radius) to be completely open, and the displacement of the pawl (*d_P_*) is 137.8 μm (error of 7.8%). The highest temperature (1160 K) occurs in the center of the V-shape silicon beam, and the maximum stress (0.66 GPa) occurs at the ends of the actuator and the soft beam, which is sufficiently safe for the melting point and breaking strength of silicon.

## 3. Fabrication

SOI wafers (50 μm device layer, 3 μm buried layer, 400 μm handle layer) are used to fabricate movable structures in the S&A device. The movable components are placed on the device layer with chambers in the handle layer for releasing moving components. The fabrication process is shown in Figure 3a, as follows: (1) Sputtering 50 nm Cr and 300 nm Au on the top of the SOI as the electrodes; (2) Spinning photoresist on both sides for masking in the Deep Reactive Ion Etching (DRIE) process; (3) DRIE process etches the bottom silicon with a depth of 400 μm to release moving parts; (4) DRIE process etches the top silicon with a depth of 50 μm; (5) Clearing the photoresist; (6) Using HF wet etching to etch the SiO_2_ for the release process, and separate the movable parts of the top silicon and the fixed parts of the bottom silicon. The two layers of the S&A device can be fabricated in one layer of the SOI wafer, as shown in Figure 3b. With the top layer covering the bottom one, the two layers are connected by epoxy resin glue, and the double-layer barrier is formed in the center of the S&A device, as shown in Figure 3c.

## 4. Tests and Discussions

### 4.1. Test of the Actuators’ Performance

The key indicators of the actuator are static performance and dynamic performance. According to the driving principle, the displacement of the electrothermal actuator is directly determined by the applied voltage, and the displacement when the applied voltage lasts for 20 ms is chosen as the static performance of the actuator, and the response process when the applied voltage lasts for 10 ms is chosen as the dynamic performance. In static performance test, the displacement of the barriers and the pawls are recorded by a high-speed camera (1000 fps) from 6 V to 13 V, and test results are shown in Figure 4a,b. Only when the displacement of the barriers and the pawls are all larger than the arming position can the S&A device switch to the arming condition. Thus, the minimum applied voltage is 12 V with the barriers’ displacement of 321.4 μm (theoretical error of 14%; simulation error of 1.2%), and the pawls’ displacement of 127.8 μm (theoretical error of 17%; simulation error of 7.8%). The simulation result is closer to the test result due to its more specific thermal conduction model.

The response process of the barrier and the pawl are recorded by a high-speed camera (5000 fps) from 12 V to 18 V within 10 ms, as shown in Figure 4c,d. With the increase of the applied voltage, the pawl and the barrier gain a faster response, and the time required to reach the arming position is greatly reduced. When the applied voltage is larger than 14 V, the barrier reaches the limit (avoid useless excessive displacement). When the applied voltage is set to 17 V, the pawl needs 2 ms to reach the arming position, and the barrier needs 3 ms, which is fast enough to switch the S&A device’s condition. However, when the applied voltage is further set to 19 V, the actuator will overheat and damage before the test time (10 ms).

### 4.2. Test of the Operation Process with Constant-Voltage Driving

According to the test results of the dynamic performance, the operation process can be designed in detail. When the applied voltage is set to 17 V, the detailed operation process is shown in Figure 5. To gain the minimum time, all steps should be started immediately when the movement cannot interfere. Therefore, the safety to arming (*S-A*) process can be divided into the following four steps, as shown in Figure 5a,c. Step 1 (*T*_1_ = 0.4 ms): Opening the pawls to release the barriers’ movement. Step 2 (*T*_2_ = 2 ms): Opening the barriers to the arming position. Step 3 (*T*_3_ = 3.4 ms): Closing the pawls to prevent the barriers from recovering to the safety position. Step 4 (*T*_4_ = 2.8 ms): Closing the barriers to engage with the pawls. All the steps of the *S-A* process can be finished within 8.6 ms with the applied voltage of 17 V. According to the reverse sequence, the S&A device can recover to the safety condition. Thus, the arming to safety (*A-S*) process can also be divided into four steps, as shown in Figure 5b,d. Step 1 (*T*_5_ = 2.2 ms): Opening the barriers to release the pawls’ movement. Step 2 (*T*_6_ = 1 ms): Opening the pawls to make the barriers can recover to the safety position. Step 3 (*T*_7_ = 2 ms): Closing the barriers to the safety position. Step 4 (*T*_8_ = 12.8 ms): Closing the pawls to the initial position. The *A-S* process can be finished within 18 ms with the applied voltage of 17 V. For clarity, Figure 5c,d only show the bottom layer of the S&A device, and the operation process is also suitable for the double-layer structure.

With the increase of applied voltage, the pawls and the barriers gain a faster response. Thus, the relationship between the applied voltage and the operation process needs to be further discussed. The tests are carried out in 13 groups (0.5 V interval from 12 V to 18 V), and the test results are shown in Figure 6. The heating steps (*T*_1_, *T*_2_, *T*_5_, and *T*_6_) are obviously shorter with the increase of the applied voltage. The displacement of the actuator is directly controlled by the temperature. Thus, the time of the cooling steps (*T*_3_, *T*_4_, *T*_7_, and *T*_8_) is independent of the applied voltage and depends on the position where the cooling starts and the target position. Therefore, the cooling steps change only slightly with the increase of the applied voltage. The specific analysis of the *S-A* processes and the *A-S* processes are as follows:

The test results of the *S-A* process are shown in Figure 6a,c. When an urgent ignition is needed, the barrier should directly open the fire hole after *T*_1_ time instead of waiting for the finish of all steps. However, *T*_1_ is nearly decreased to zero (0.3 ms with the applied voltage of 18 V), which means that the barrier can be driven to the arming position almost immediately with a high applied voltage. *T*_2_ obviously decreases from 13 ms to 1.9 ms, which is consistent with the test results of dynamic performance. *T*_3_ slightly decreases (5.4 ms to 3.2 ms) for the decrease of *T*_1_ and *T*_2_. However, in the *T*_3_ process, the barrier should stay in the arming position to wait for the pawl, which will cause the barrier actuator to overheat and increase *T*_4_ slightly (1.4 ms to 2.9 ms). In general, the total time of the *S-A* process (*T*_1_ + *T*_2_ + *T*_3_ + *T*_4_) obviously decreases from 22.2 ms to 8.3 ms, the trend of which is basically consistent with the barrier-driven time (*T*_2_ + *T*_3_). When the applied voltage is 18 V, the cooling process (*T*_3_ + *T*_4_) is 6.1 ms (73.5% of the total time). Thus, the *S-A* process is hardly faster with the further increase of the applied voltage.

The *A-S* process is similar to the *S-A* process, and the test results are shown in Figure 6b,d. *T*_5_ obviously decreases from 13.6 ms to 2 ms, and *T*_6_ decreases from 8 ms to 1 ms, which is consistent with the test results of dynamic performance. With the decrease of *T*_5_ and *T*_6_, *T*_7_ also slightly decreases from 2.4 ms to 1.8 ms, and *T*_8_ decreases from 13.4 ms to 12.6 ms. Therefore, the total time of the *A-S* process (*T*_5_ + *T*_6_ + *T*_7_ + *T*_8_) also obviously decreases from 37.4 ms to 17.4 ms, the trend of which is basically consistent with the barrier-driven time (*T*_5_ + *T*_6_). When the applied voltage is 18 V, the cooling process (*T*_7_ + *T*_8_) is 14.4 ms (82.8% of the total time). Thus, the *A-S* process is also hardly faster with the further increase of the applied voltage.

### 4.3. Test of the Operation Process with Capacitor–Discharge Driving

According to the above discussion, the eight steps of the two processes (the *S-A* process and the *A-S* process) vary with the applied voltage, and the electric energy required in the two processes can be calculated by applying the following equations:(4)$$\left\{\begin{array}{l} E_{S-A}=V_{IN}{}^{2}(\frac{T_{1}+T_{2}}{R_{P}}+\frac{T_{2}+T_{3}}{R_{B}})\\ E_{A-S}=V_{IN}{}^{2}(\frac{T_{6}+T_{7}}{R_{P}}+\frac{T_{5}+T_{6}}{R_{B}}) \end{array}\right.$$

Here, *E_S–A_* is the electric energy required in the *S*–*A* process. *E_A–S_* is the electric energy required in the *A*–*S* process. *V_IN_* is the applied voltage. *R_P_* and *R_B_* are the resistances of the pawl actuators and the barrier actuators. The calculation results are shown in Figure 7a. Because the two processes are hardly faster with the further increase of the applied voltage, both *E_S–A_* and *E_A–S_* have a minimum value when the applied voltage is set to 17 V, and *E_A–S_* is always less than *E_S–A_*.

With the introduction of the bistable mechanism, the S&A device can switch the condition with limited energy. Thus, the two processes can also be carried out by the discharge of a capacitor, and the drive circuit is shown in Figure 7b. *R*_1_ is used to limit the charging current of the capacitor, and the operation process is controlled by two MOSFETs. Considering the influence of *R*_1_, the discharge process can be obtained by the following equations:(5)$$\left\{\begin{array}{l} V_{0}=V_{IN}\frac{e^{-\frac{t}{\tau }}+\frac{R_{2}}{R_{1}}}{1+\frac{R_{2}}{R_{1}}}\\ \tau =\frac{R_{2}C}{1+\frac{R_{2}}{R_{1}}} \end{array}\right.$$

Here, *τ* is the time constant. *R*_2_ varies with the steps, which is *R_P_* (*T*_1_ and *T*_7_), *R_B_* (*T*_3_ and *T*_5_), or *R_P_ R_B_*/(*R_P_* + *R_B_*) (*T*_2_ and *T*_6_). When *R*_1_ is much higher than *R*_2_, the influence of *R*_1_ is negligible. A tantalum capacitor of 2000 μF with a charging voltage of 17 V is used to drive the S&A device, and the test results are recorded by an oscilloscope and shown in Figure 7c,d. The capacitor voltage (*V*_0_) decreases to 12.54 V in the *S-A* process and 13.22 V in the *A-S* process, which is higher than the minimum applied voltage of 12 V. When the charging voltage is less than 16.5 V, the electric energy of the capacitor is insufficient to drive the S&A device. The total time of the S–A process is 9.6 ms and the A–S process is 20.8 ms, which is slightly longer than the constant-voltage driving with the applied voltage of 17 V. However, confined by the *R*_1_ of 100 Ω in the capacitor–discharge driving, the S&A device needs an interval of 1 s between two operation processes to recharge the capacitor. Although the capacitor–discharge driving is slower than the constant-voltage driving, it reduces the requirement for the power supply of the weapon system and widens the application scenario.

## 5. Conclusions

(1)In this paper, we propose the design and characterization of a double-layer electrothermal MEMS S&A device with a bistable mechanism. Four groups of the bistable mechanism in the S&A device can effectively improve the safety and reliability of the weapon system. Each bistable mechanism drives a semi-circular barrier and a pawl independently through two V-shape electrothermal actuators and controls their engagement through a specific operation process. The four groups of the bistable mechanism are axisymmetrically placed in two layers, two mechanisms for each layer, to constitute a double-layer barrier structure in the center of the S&A device. When a voltage is applied on the actuators, the barriers and the pawls can be driven to an arming position. The test results of the actuators’ static performance show that the applied voltage is at least 12 V to drive the S&A device. The test results of the dynamic performance show that the actuators gain a faster response with the increase of the applied voltage.(2)Only according to a specific operation process can the S&A device switch the conditions between the safety and the arming. The arming to safety process is the inverse of the safety to arming process, and all of them have four steps. Due to the faster response, the total time of the two processes decreases with the increase of applied voltage, but the two processes are hardly faster when the applied voltage is more than 18 V. When the applied voltage is set at 17 V, the S&A device can use minimum electric energy to switch the condition to the arming within 8.6 ms or the safety within 18 ms. Besides the constant-voltage driving, the S&A device can also switch the condition through the discharge of a capacitor of 2000 μF. When the charging voltage is set at 17 V, the S&A device can switch the condition to the arming within 9.6 ms or the safety within 20.8 ms. Thus, the test results show that the S&A device can use constant-voltage driving or the capacitor–discharge driving to drive the double-layer barrier to the safety or the arming position and keep it on the position passively by the bistable mechanism.

## Figures and Tables

**Figure 1 micromachines-13-01076-f001:**
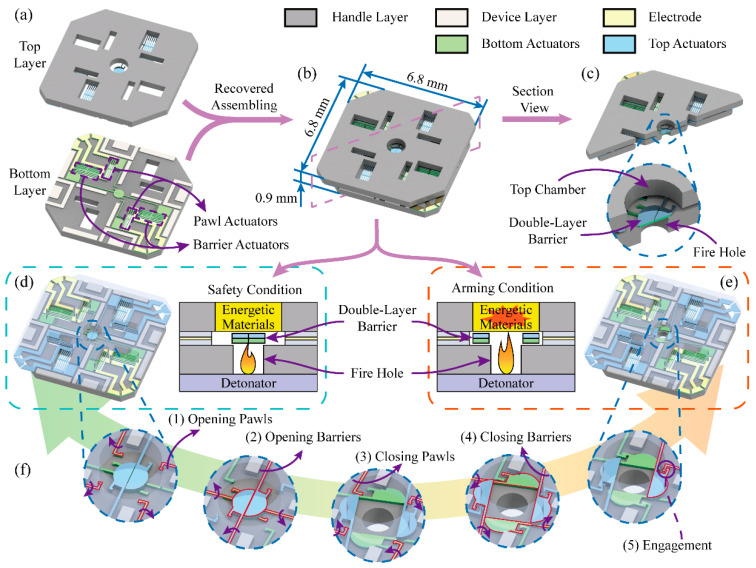
Fundamental mechanism of the S&A device: (**a**) Two layers structure; (**b**) The S&A device; (**c**) The section view; (**d**) The safety condition; (**e**) The arming condition; (**f**) Operation process of the bistable mechanism.

**Figure 2 micromachines-13-01076-f002:**
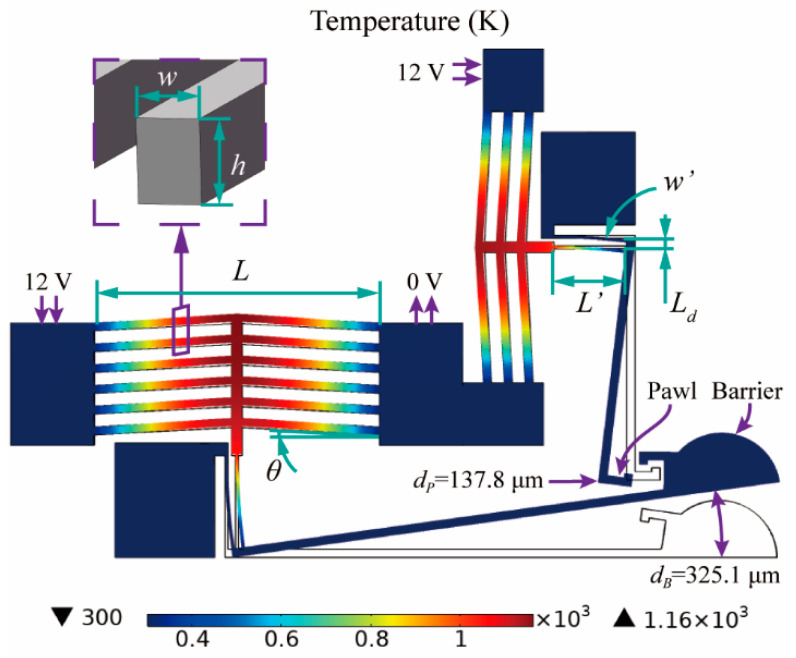
Simulation results of the electro-thermal actuator.

**Figure 3 micromachines-13-01076-f003:**
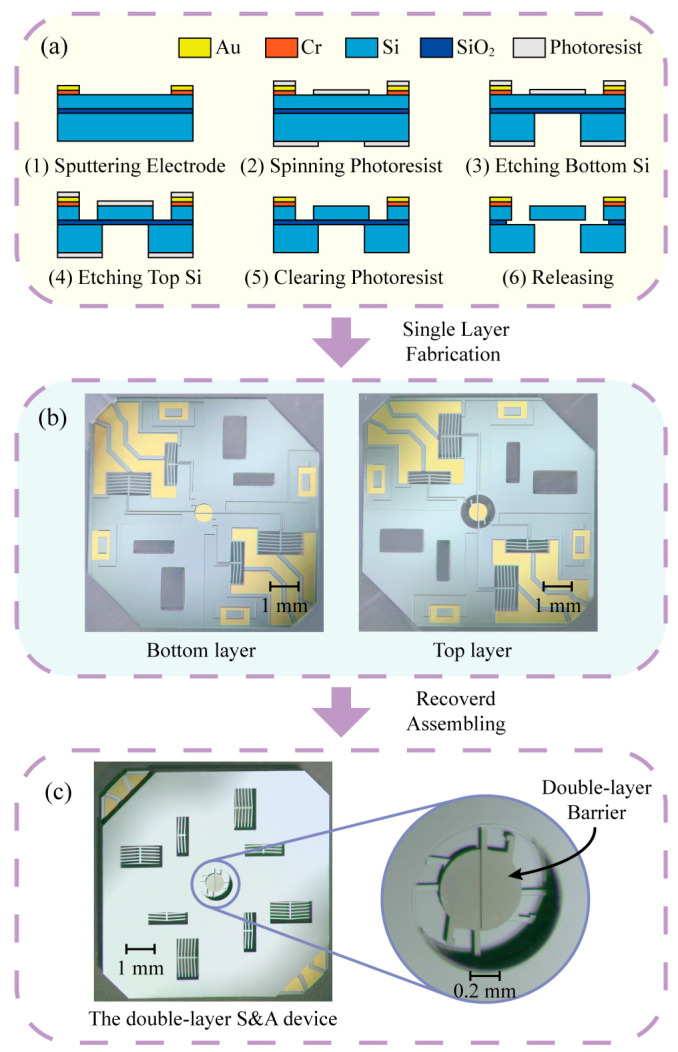
Fabrication process for the S&A device: (**a**) Fabrication process; (**b**) Two layers of the S&A device; (**c**) The double-layer S&A device.

**Figure 4 micromachines-13-01076-f004:**
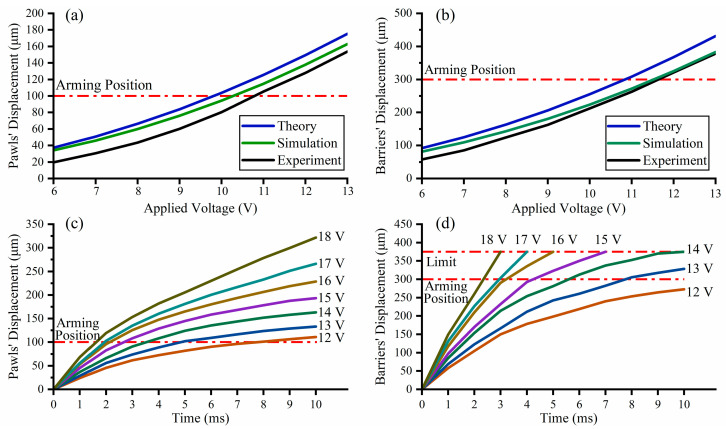
Test results of the actuators’ performance: (**a**) Static performance test results of the pawl actuators; (**b**) Static performance test results of the barrier actuators; (**c**) Dynamic performance test results of the pawl actuators; (**d**) Dynamic performance test results of the barrier actuators.

**Figure 5 micromachines-13-01076-f005:**
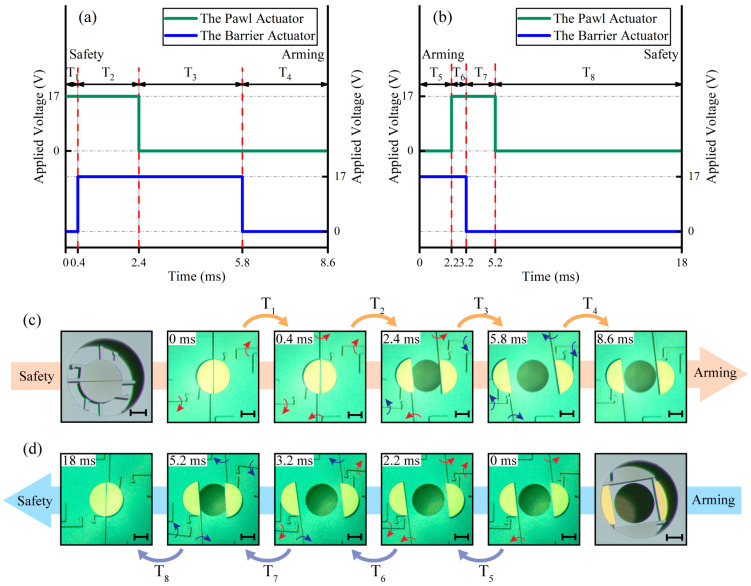
Test results of the operation process: (**a**) The safety to arming process; (**b**) The arming to safety process; (**c**) The safety to arming process; (**d**) The arming to safety process (Scale bar 0.2 mm).

**Figure 6 micromachines-13-01076-f006:**
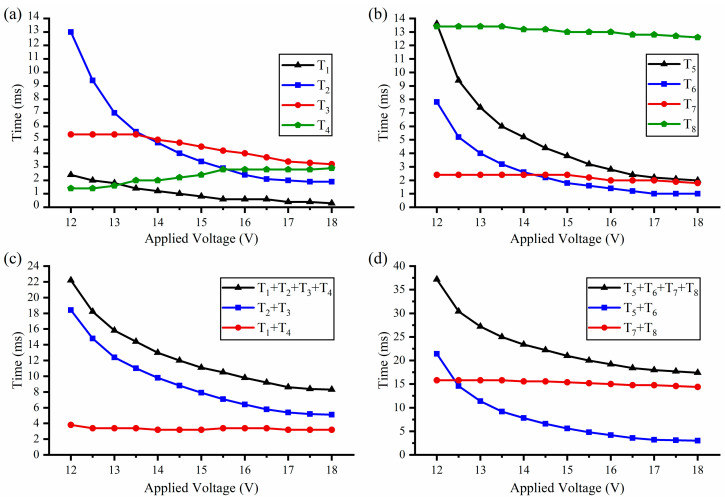
The relationship between the applied voltage and the operation process: (**a**) The safety to arming process; (**b**) The arming to safety process; (**c**) The safety to arming process; (**d**) The arming to safety process.

**Figure 7 micromachines-13-01076-f007:**
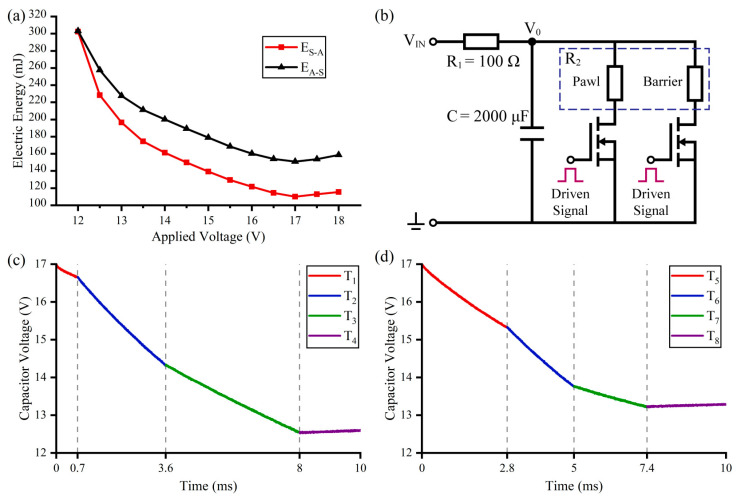
Test results of the operation process with capacitor–discharge driving: (**a**) The electric energy required in the constant-voltage driving; (**b**) The drive circuit; (**c**) The safety to arming process; (**d**) The arming to safety process.

**Table 1 micromachines-13-01076-t001:** Geometrical parameters of the electro-thermal actuator.

Item	Width (*w*)	Length (*L*)	Thickness (*h*)	Angle(*θ*)	Number of Beams
Barrier Actuator	38	1370	50	3	6
Pawl Actuator	38	1305	50	3	3
Unit	μm	μm	μm	°	None

**Table 2 micromachines-13-01076-t002:** Geometrical parameters of the soft lever mechanism.

Item	Width (*w’*)	Length (*L’*)	Distance between Two of the Soft Beams (*L_d_*)	Enlarged Proportion
Barrier Actuator	14	450	100	47.44
Pawl Actuator	14	350	100	23.26
Unit	μm	μm	μm	None

## Data Availability

Not applicable.

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
