# Peer review of "Design of a Double-Layer Electrothermal MEMS Safety and Arming Device with a Bistable Mechanism"

_micromachines, 2022, doi:10.3390/mi13071076_

Round 1

Reviewer 1 Report

It's a nice device and very well explained.  Would be interesting for others designing similar mechanisms for this or for other applications.  I think it would be interesting to look at how you could develop a system where it was much easier to disarm (safety) rather than arm, or which would "fail-safe".  I think what's maybe also missing with such a concept is research on the reliability of MEMS S&A devices, such as repeated actuation, failure rates etc. but that's probably several additional publications.  

Reviewer 2 Report

This paper did a great work on the SA device. The safety and arming state can be well maintained by the locking of pawls. The paper can be accepted for publication, but some revisions are still needed to improve the paper.

 1. The figures show a low resolution and lose important information.

a) The structure for pawls/barriers and actuators are not clearly shown in Figure 1.

b) The initial state before arming are not shown in Fig.1f.

c) The quality of other figures should be improved.

2. The English writing should be improved.

a) Many expressions are confusing.

e.g.1 In S2.1 “(4) Recovering the barriers to engage with the pawls”. Why recover the barriers in arming sate? The authors did not means move barriers back to the initial state, but they used “recover”.

 e.g. 2. In S4.2. “Step 1 (T5 = 2.2 ms): Opening the barriers to release the movement of the pawls. Step 2 (T6 = 1ms): Opening the pawls to release the movement of the barriers.” In Step 2 the barriers are released but the barriers should be moved in Step 1. Though, I get the real process, but these words cannot clearly show the process.

b) All the paper should be fully revised to improve the English writing. It may be a hard work to read many of the sentences.

3. The conclusion said “The test results of the actuators’ static performance are basically consistent with the theoretical and the simulation results”.  However, an error up to 17% is reported.  The word “consistent” cannot get a solid basis. An explanation about the division should be added in the paper.

4. About the Introduction.

a) Bistable mechanism has been used in switches. So it is an important to distinguish your work by clearly show the reasons for why the existing methods cannot work in SA devices. Only small displacement?

b) A outline for the whole paper should be added at the end of Introduction.

 5. The references should be checked. Many paper cannot get their whole information.

Reviewer 3 Report

The description of the research activity provided in this manuscript looks of an incremental nature, based on much the same authors have probably done in past activities. The paper is therefore quite poor in the description of some fundamental parts of the activities themselves, and far from being self-contained. In the reviewer’s opinion, it cannot therefore be accepted unless the authors provide major revisions according to the following:

1.       Stating in the titles something like “research on…” looks rather obvious: if it were no research, why would it then to be published in a scholarly journal? The title needs then to be rephrased.

2.       In the keyword, the authors used bistable. Bistable what?

3.       The introduction is poor in content: why does this work? what’s missed in the literature? what’s the real goal of this research? what’s the target in terms of device efficiency/performance? what is the structure of the current manuscript?

4.       Section 2: starting from here, it is said that the device is made of two layers: why? what’s the benefit of it? what are the major problems encountered in microfabrication? The top and bottom are linked to a rather simple description: the process to get this would probably to be given first.

5.       Where is bi-stability described in details? Why it? What’s the benefit of some other working mechanisms?

6.       Regarding fig.1, the description does not allow to understand what the device has been designed to do.

7.       Section 2.2: in the description of the variables in the equations, better to use a comma in the list.

8.       In Eq. 1, what is x? A sketch needs to be added to show all the details much better. How is the angle theta in Table 1 accounted for by x?

9.       Table 1: what is “number of the beam”? It is maybe the “number of beams”?

10.   At page 4, what is the “bending deformation”? Show an enlarged view of the deformed shape to understand it.

11.   Several equations are included without any reference to the literature, which probably means that they are discovered here for the first time (which is hard to believe).

12.   Page 4: what is Young’s modulus of Si adopted in the analyses? We all know that Si has an FCC structure, so it is not isotropic.

13.   What about the reference temperatures of more than 1000 K. How is it handled in practice? How does it compare to the melting point of everything in the device? Is it the device obtained from a SoI wafer: if such, how is the oxide affected by such temperatures?

14.   Gpa is to be GPa.

15.   Is the reported values of the “maximum stress” really mesh-independent? Is it the max principal stress or what?

16.   Fig. 2 would probably need also a sketch to understand the behavior of the device.

17.   Beginning of section 4.1: what does “static performance and dynamic performance” mean? This introductory part needs a better description.

18.   At the end of page 6 authors refer to a theoretical error: which theory has been adopted? where is it described?

19.   Fig. 4: we guess readers would like to have an explanation regarding what the Pawls and Barriers displacements are.

20.   Page 7: the phrasing “response speed of heat transfer” is totally meaningless.

21.   Again at page 7, stating “to gain the fastest speed” is grammatically and concretely still with no meaning.

22.   Why is so always important to discuss the speed, to have the device “faster” (than what)?

23.   Fig. 6 needs to be better described too.

24.   At the end of section 4 there are no in-depth discussions to understand if the goal/aims of the research activities have been achieved.

25.   The format of the references does not in compliance with the journal one.

Round 2

Reviewer 2 Report

The revision greatly improves the paper. The paper can be accepted now. A minor revision is needed about the reason for the division between the simulated and experimental result. The words “more specific thermal conduction model.”  are not strong enough. A more detailed description is needed.

Reviewer 3 Report

I do not think the authors have understood that my comments were asking for the manuscript to be improved, and not to provide me with some "answers".